# Genetic compatibility and ecological connectivity drive the dissemination of antibiotic resistance genes

David Lund [1,2], Marcos Parras-Moltó[1,2], Juan S. Inda-Díaz [1,2], Stefan Ebmeyer[2,3], D. G. Joakim Larsson [2,3], Anna Johnning [1,2,4] & Erik Kristiansson [1,2] ✉

The dissemination of mobile antibiotic resistance genes (ARGs) via horizontal gene transfer is a significant threat to public health globally. The flow of ARGs into and between pathogens, however, remains poorly understood, limiting our ability to develop strategies for managing the antibiotic resistance crisis. Therefore, we aim to identify genetic and ecological factors that are fundamental for successful horizontal ARG transfer. We used a phylogenetic method to identify instances of horizontal ARG transfer in ~1 million bacterial genomes. This data was then integrated with >20,000 metagenomes representing animal, human, soil, water, and wastewater microbiomes to develop random forest models that can reliably predict horizontal ARG transfer between bacteria. Our results suggest that genetic incompatibility, measured as nucleotide composition dissimilarity, negatively influences the likelihood of transfer of ARGs between evolutionarily divergent bacteria. Conversely, environmental co-occurrence increases the likelihood, especially in humans and wastewater, in which several environment-specific dissemination patterns are observed. This study provides data-driven ways to predict the spread of ARGs and provides insights into the mechanisms governing this evolutionary process.

The prevalence of antibiotic-resistant pathogens keeps increasing, threatening to drastically reduce our ability to efficiently treat and prevent bacterial infections[1]. Bacteria can develop resistance by acquiring mobile antibiotic resistance genes (ARGs) through horizontal gene transfer (HGT)[2,3]. Given sufficient antibiotic selection pressure, HGT allows for ARGs to successfully become established even among evolutionarily distant members of the bacterial community[4]. In contrast to the development of drug resistance through chromosomal mutations, HGT allows larger genetic regions—encompassing multiple ARGs providing resistance to several antibiotics—to be transmitted in a single event[5]. The horizontal transfer of ARGs is, thus, fundamental for the rapid evolution of multidrug-

resistant pathogens, constituting a growing threat to human health globally[6].

A wide range of mobile ARGs have been characterized to date. As an example, the ResFinder database currently includes over 2500 gene sequences linked to resistance to a total of 17 classes of antimicrobial compounds[7]. Most of these genes were discovered only after pathogens or pathobionts acquired them. Many of the ARGs carried by pathogens, however, are hypothesized to originate from non-pathogenic species in external or host-associated environments[8–10]. Indeed, recent studies have demonstrated that environmental and commensal bacteria maintain an extensive collection of ARGs that can be mobilized and, eventually, transferred into pathogens[11–13]. This

[1]Department of Mathematical Sciences, Chalmers University of Technology and University of Gothenburg, Gothenburg, Sweden. [2]Centre for Antibiotic Resistance Research in Gothenburg (CARe), University of Gothenburg, Gothenburg, Sweden. [3]Department of Infectious Diseases, Institute of Biomedicine, Sahlgrenska Academy, University of Gothenburg, Gothenburg, Sweden. [4]Department of Systems and Data Analysis, Fraunhofer-Chalmers Centre, Gothenburg, Sweden. ✉e-mail: erik.kristiansson@chalmers.se

fundamental process, where ARGs are transferred from evolutionarily distant bacteria—potentially via several intermediate hosts—into a pathogen, remains poorly understood.

Horizontal gene transfer is the intricate process of acquisition and, potentially, chromosomal integration of exogenous genetic material[14]. The transfer of ARGs is primarily mediated via conjugation, where plasmids and other conjugative elements are transferred via pili, and transformation, which involves the direct uptake of DNA from the environment. Transduction, i.e. the transfer of genetic material via bacteriophages, as well as transmission via membrane vesicles, may also play a role[14-17]. Even though the medical consequences of the horizontal transfer of ARGs are clear[18], we lack fundamental insights into the factors that govern how ARGs are transferred and, thus, in what pathogenic and non-pathogenic bacteria they will eventually appear. In particular, successful conjugation and transformation are highly dependent on the genetics of the involved cells, however, so far most studies have focused on the presence of mobile genetic elements or other biochemical functions[19,20]. The importance of other vital parameters, such as the genetic composition of the acquired ARGs and the recipients' genomes has received much less attention. Furthermore, the influence of ecological factors such as connectivity[21,22] and, accordingly, in what bacterial communities the transfers are most likely to happen, remains unclear[23-25]. Consequently, we are unable to forecast how ARGs are disseminated, which is central for developing appropriate countermeasures to limit the spread of emerging ARGs— and thus to preserve the potency of existing and future antibiotics.

In this study, we aimed to address these knowledge gaps and identify genetic and ecological factors governing the horizontal transfer of ARGs. We developed a machine learning-based approach—leveraging over 2.6 million ARGs identified in almost 1 million genomes—able to accurately predict horizontal ARG transfer between bacterial hosts. Our results show that genetic incompatibility, both between the hosts' genomes and between the acquired gene and the recipient's genome, strongly limits the transfer of ARGs, especially between evolutionary distant bacteria. Furthermore, we demonstrate that ecological connectivity, measured as the co-occurrence patterns from >20,000 metagenomes, significantly facilitates the spread of ARGs. Our results, particularly, support that ARG transfers deducible from current genome sequence repositories have predominantly happened in the human and wastewater microbiomes. This study, thus, provides data-driven ways to predict the dissemination of ARGs and reveals insights into the factors that govern how resistance genes spread between bacterial hosts.

## Results

### Identification of horizontally spread resistance genes

A total of 867,318 bacterial genomes (Supplementary Data 1) were screened for 22 ARG classes representing ten resistance mechanisms. Briefly, these mechanisms included two types of aminoglycoside-modifying enzymes; aminoglycoside acetyltransferases (AAC) and aminoglycoside modifying phosphotransferases (APH), two categories of beta-lactamases; non-metallo-beta-lactamases (class A, C, D), and metallo-beta-lactamases (class B), two types of macrolide resistance; Erm 23S rRNA methyltransferases and Mph 2′-macrolide phosphotransferases, three types of tetracycline resistance; tetracycline efflux pumps (Tet efflux), tetracycline inactivating enzymes (Tet enzyme), and tetracycline ribosomal protection genes (Tet RPG), and quinolone resistance genes (Qnr) (Figs. 1, 2). In total, 2,666,002 matching ARGs encoding 60,773 unique protein sequences were found, among which aminoglycoside and beta-lactam resistance genes dominated (40.9% and 38.4% of matches, respectively; Fig. 2a, Supplementary Table 1). A phylogenetics-based method was used to search for horizontal transfer of ARGs between distantly related bacterial species. Briefly, phylogenetic trees were constructed using the translated ARGs for each ARG class, and in each tree, successful transfers were identified based on nodes with descendants representing highly similar ARGs carried by hosts with at least an order-level taxonomic difference (Supplementary Fig. 1). For each observed transfer, we compiled information on the genetic incompatibility of the involved ARGs and host genomes (measured by the difference in nucleotide composition and genome size), the co-occurrence of the hosts in bacterial communities based on 20,816 metagenomes from five different environment types, and the type of host cell envelopes. For full details, see "Methods".

In total, 6276 horizontal transfers of ARGs were identified (Fig. 2b), where transfers involving aminoglycoside phosphotransferases (APHs) and class A, C, or D beta-lactamases were most common (29.9% and 23.8%, respectively). The number of transfers generally increased with the number of predicted ARGs encoding each resistance mechanism, except for aminoglycoside acetyltransferases (AACs) and class B beta-lactamases, which were both significantly underrepresented among the identified transfers ($p < 0.01$, two-sided Fisher's exact test, Supplementary Fig. 2). In the identified transfers, the similarity of the ARGs carried by the evolutionarily distant hosts varied. The majority of APHs, Erm 23S rRNA methyltransferases, Mph macrolide 2′-phosphotransferases, tetracycline efflux pumps, and tetracycline ribosomal protection genes (RPGs) had an amino acid

**Fig. 1 | Overview of the analysis pipeline.** First, bacterial genomes were screened for antibiotic resistance genes (ARGs). Second, phylogenetic trees were constructed using the identified protein sequences, and horizontal transfer was inferred from the trees by detecting similar genes carried by evolutionarily distant hosts. For each identified instance of horizontal transfer, data describing genetic incompatibility and co-occurrence in bacterial communities was collected. Here, genetic incompatibility was estimated as the nucleotide composition dissimilarity between the involved genomes and their ARGs, while environmental co-occurrence was estimated by mapping the involved genomes onto a large metagenomic dataset. Finally, the collected data was used to train random forest models. Factors influential for predicting horizontal transfer were identified using feature importance analysis.

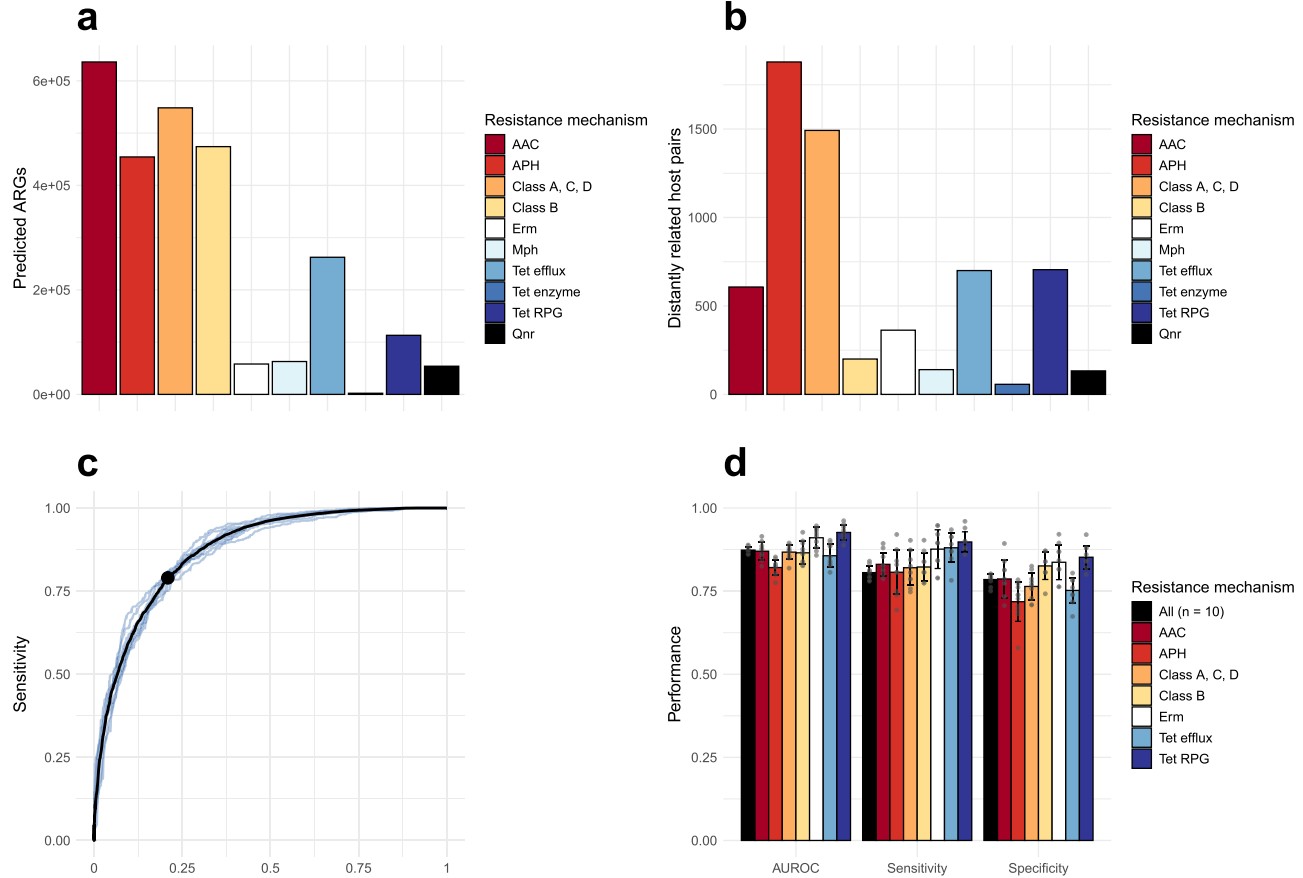

**Fig. 2 | Summary of identified horizontal transfers of antibiotic resistance genes (ARGs), and performance of random forest models trained to predict horizontal ARG transfer. a** Total number of ARGs predicted in 867,318 bacterial genomes, stratified based on encoded resistance mechanism. Included among the resistance mechanisms are aminoglycoside acetyltransferases (AAC), aminoglycoside phosphotransferases (APH), class A, C, D beta-lactamases, class B beta-lactamases, Erm 23S rRNA methyltransferases, Mph 2'-macrolide phosphotransferases, tetracycline efflux pumps (Tet efflux), tetracycline inactivating enzymes (Tet enzyme), tetracycline ribosomal protection genes (Tet RPG), and quinolone resistance genes (Qnr). **b** Total number of detected instances of ARGs horizontally transferred between distantly related bacterial hosts, stratified based on encoded resistance mechanism. **c** Receiver operating characteristic curves produced from predictions on test data by random forest models trained on horizontal transfers representing all included resistance mechanisms, over ten iterations. Each model was built using variables representing the genetic incompatibility, environmental co-occurrence, and cell envelope of the bacteria involved in each transfer. The black line represents the mean of the produced receiver operating characteristics (ROC) curves. The point represents the mean optimal performance (the point closest to a sensitivity and specificity of 1). **d** Area under the ROC curve (AUROC), sensitivity, and specificity observed for predictions on test data using random forest models representing different resistance mechanisms with enough data present (>100 transfers observed). The bars show the mean +/− SD of the observed metrics over ten iterations. Source data are provided as a Source Data file.

identify >99% between gene variants, suggesting more recent transfers. In contrast, transfers of AACs, class A, C, and D beta-lactamases, and, especially, class B beta-lactamases and tetracycline-inactivating enzymes included a larger proportion of genes with a lower sequence similarity (Supplementary Fig. 3).

**Random forests can accurately predict the horizontal transfer of ARGs**

A machine learning model using a random forest was created to predict the horizontal transfer of ARGs between bacterial hosts. The predictions were based on genetic incompatibility, environmental co-occurrence, the Gram staining properties of the host pairs, as well as the gene class of the transferred ARG. Briefly, factors representing genetic incompatibility included the mean nucleotide composition dissimilarity between the distantly related host genomes (genome 5 mer distance), the maximal observed nucleotide composition dissimilarity between the transferred ARG and an involved host genome (gene-genome 5 mer distance), as well as the proportional difference in mean size between the distantly related host genomes. Co-occurrence was estimated in five different types of environments (animal, human,

soil, water, and wastewater) by mapping the bacterial genomes onto a large metagenomic dataset and calculating the proportion of samples where both distantly related hosts were present. Finally, Gram staining was determined based on the phyla of the bacteria involved (for full details, see "Methods"). First, a general model was trained using the transfers of all included ARG classes. In addition, seven mechanism-specific models were trained by stratifying the transfers based on the resistance mechanism encoded by the transferred ARGs. All models were trained using a positive dataset, containing observed transfers, and a negative dataset, created by permuting the leaves in the ARG trees, representing the assumption that successful transfers occur randomly between bacterial genomes carrying at least one ARG (see "Methods").

Evaluation of the models showed high accuracy in predicting the horizontal transfer of ARGs between bacterial hosts, with the general model having a mean area under the receiver operating characteristic curve (AUROC) of 0.873, a mean sensitivity of 0.806, and a mean specificity of 0.785 (Fig. 2c, d). The mechanism-specific models performed similarly well, with mean AUROC values between 0.821 (APH) and 0.926 (Tet RPG) (Fig. 2d; Supplementary Fig. 4), mean sensitivities

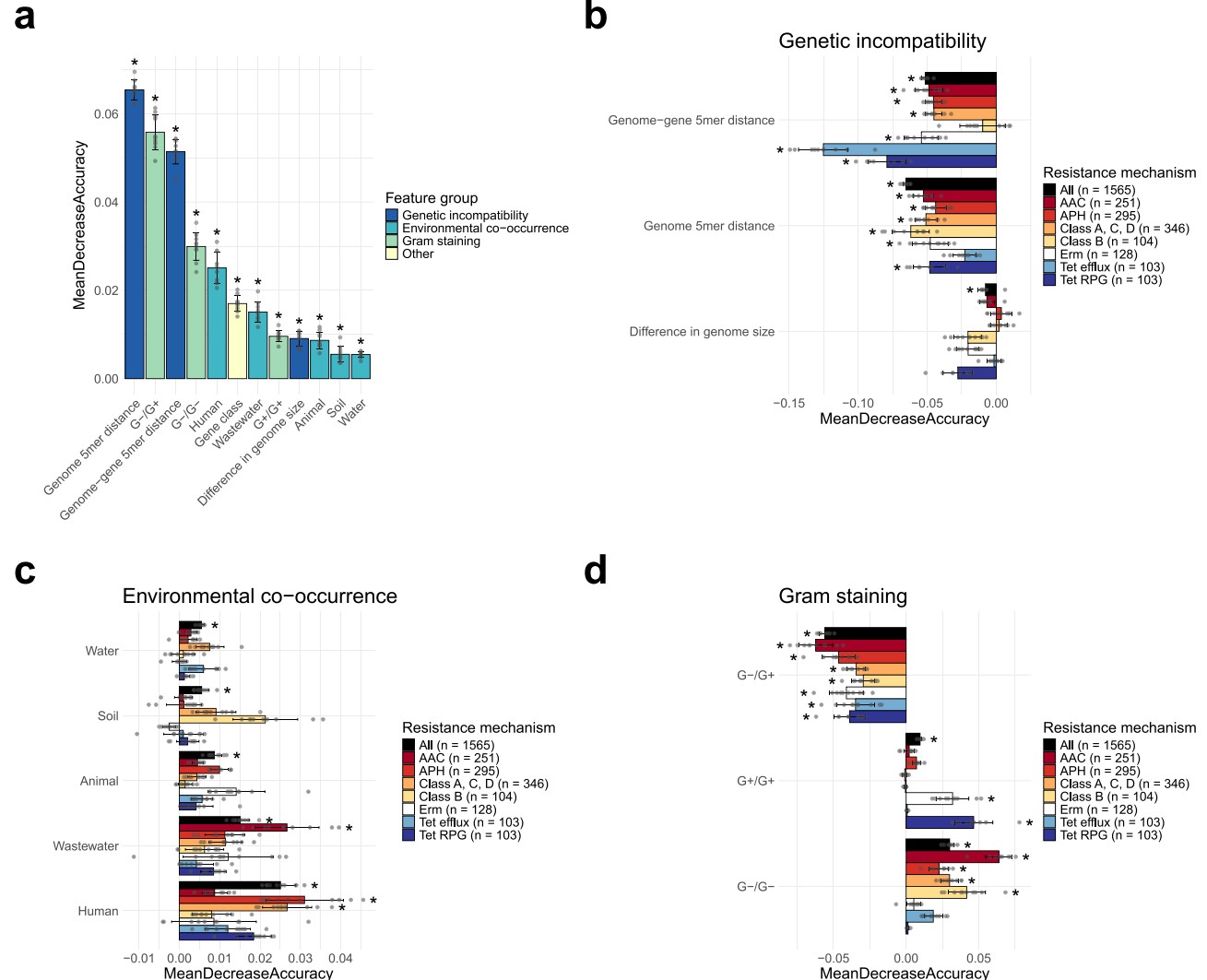

**Fig. 3 | Relative importance of genetic and environmental factors for predicting the horizontal transfer of antibiotic resistance genes.** In all instances, the bars show the mean +/− SD of the importance of each factor to the accuracy of the model (MeanDecreaseAccuracy) over ten iterations. Permutation tests were used to generate a *p*-value for each factor and iteration. All individual *p*-values are available in the Source Data. *$P < 0.01$ across all model iterations. **a** The importance of the factors included in the general random forest model, based on all observed transfers ($n = 1565$), ordered according to their overall contribution to the accuracy of the model. **b**–**d** The mean importance of each factor group (genetic incompatibility, environmental co-occurrence, and Gram staining, respectively) for each random forest models over ten iterations. In addition to the general model (All), seven models specific to different resistance mechanisms, including aminoglycoside acetyltransferases (AAC), aminoglycoside phosphotransferases (APH), class A, C, D beta-lactamases, class B beta-lactamases, Erm 23S rRNA methyltransferases, tetracycline efflux pumps (Tet efflux), and tetracycline ribosomal protection genes (Tet RPG) are included. Signs have been added to show whether an increased value of the variable is generally indicative of horizontally spread resistance genes (+) or not (−) based on partial dependence analysis. The number of observed transfers making up the training + test data for each model is included in the legends. An equal number of randomized transfers were used for each model as the negative dataset. Source data are provided as a Source Data file.

between 0.807 (APH) and 0.898 (Tet RPG) and mean specificities between 0.718 (APH) to 0.852 (Tet RPG). Thus, all models were able to accurately identify most transfers while maintaining a low false positive rate.

Next, we investigated how the different factors influenced the predictive performance of the models. This was done by permuting the response variable and calculating importance estimates for each factor (based on the mean decrease in accuracy after its removal) together with corresponding *p*-values assessing the significance (Fig. 3a, Supplementary Fig. 5)[26]. We then used partial dependence analysis[27] to assess whether each factor had a positive or negative influence on the likelihood of horizontal transfer (Fig. 3b–d).

For all models, the nucleotide composition dissimilarity between genomes and between genomes and ARGs had large and significant influences that negatively affected the likelihood of successful gene transfer (Fig. 3a, b). This effect was especially pronounced for genes encoding tetracycline efflux pumps and ribosomal protection genes (Supplementary Fig. 5). Similarly, hosts with different cell envelopes, represented by Gram staining, affected horizontal transfer negatively for all resistance mechanisms (Fig. 3d). Here, it is important to note that bacteria with different types of cell envelopes tend to use different mobile genetic elements for transferring genes horizontally, and the distance an ARG can move from its original host is generally dictated by the promiscuity of the mobile genetic elements that carry it[15]. Transfer between Gram-negatives seemed to be favored for all resistance mechanisms with the notable exception of *erm* genes and tetracycline RPGs, which were instead positively associated with transfer between Gram-positives. Comparatively, the difference in size between the host genomes was found to have a smaller, but significant, influence across all iterations of the general model (Fig. 3a, b).

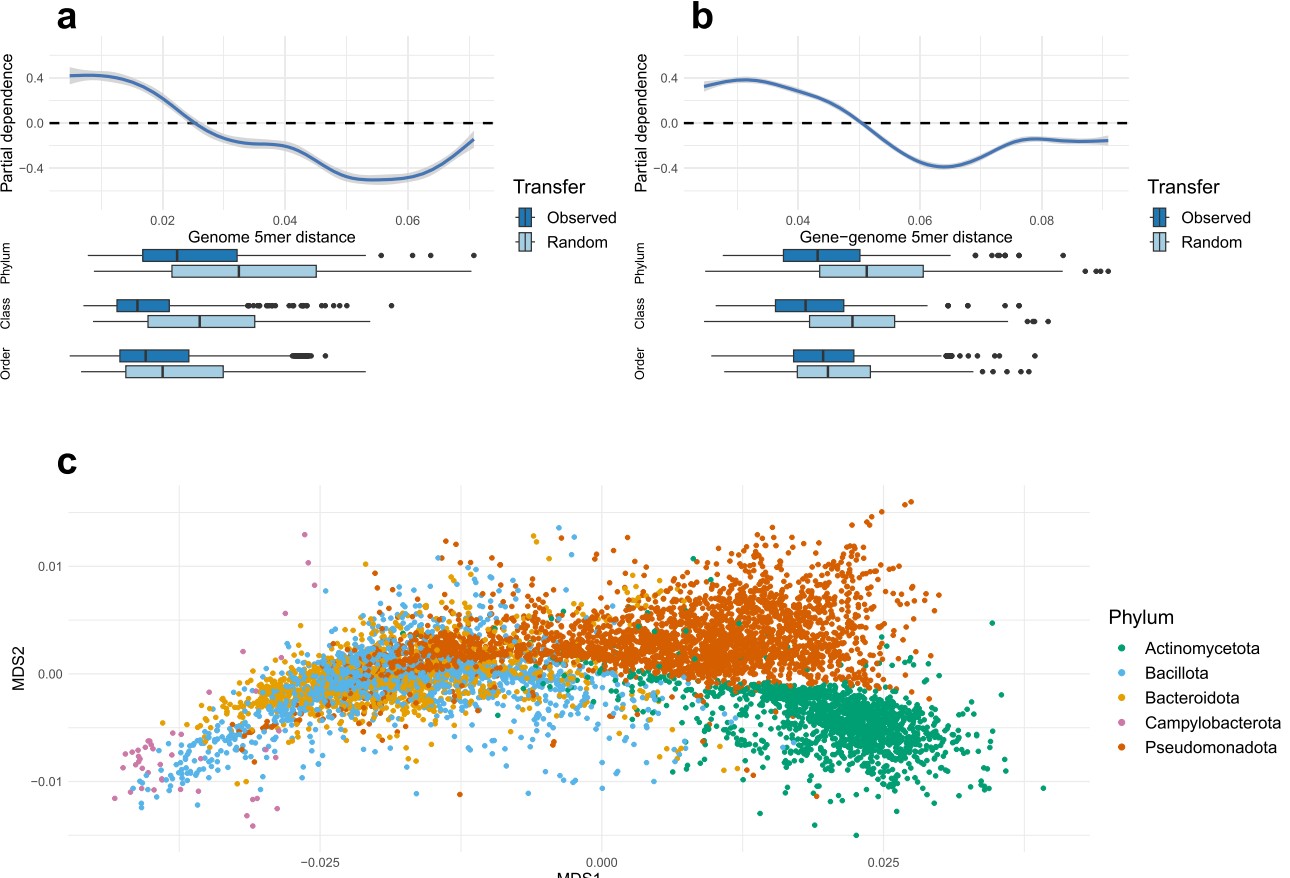

**Fig. 4 | Relative contribution of genetic incompatibility for prediction of the horizontal transfer of antibiotic resistance genes.** For each factor, the distribution of values seen for the observed and the randomized transfers at different taxonomic levels is visualized as boxplots below the main graph. Here, the centerline, box limits, and whiskers indicate the median, interquartile range, and 1,5 × interquartile range, respectively. The minima and maxima represent the minimum and maximum values observed. Only results from the general random forest model, including all resistance mechanisms, are shown. **a** Maximal 5-mer distance between antibiotic resistance gene and host genome(s). **b** Mean 5-mer distance between genome(s). **c** Multidimensional scaling, based on Euclidean distance, of the 5-mer distributions from selected genomes representing each identified host species ($n = 7609$) from the major bacterial phyla. Source data are provided as a Source Data file.

Hosts that commonly co-occurred in bacterial communities generally had a positive impact on the likelihood of horizontal transfer (Fig. 3c). Here, the human microbiome showed the highest importance in the general model, followed by wastewater and animals (Fig. 3a). A positive association was also seen for the environmental bacterial communities, but the influence was lower. In the mechanism-specific models, the highest impact of high co-occurrence on horizontal transfer was seen for APHs and class A, C, and D beta-lactamases in the human microbiome and AACs in wastewater bacterial communities (Fig. 3c).

## High genetic incompatibility decreases the transfer of antibiotic-resistance genes

To further analyze how the genetic incompatibility between host genomes and ARGs influenced the predictions of the random forest models, we generated partial dependence plots for these factors (Fig. 4a, b, Supplementary Fig. 6, 7). Briefly, these plots illustrate the marginal effect of a specific feature on the classification accuracy of the model. Positive values correspond to an increased likelihood of horizontal transfer and vice versa[27]. Our results showed that increased nucleotide composition dissimilarities reduced horizontal transfer of ARGs almost monotonically. The genetic incompatibility between the host genomes, which had the largest influence in the general model, exhibited a steep curve with a negative contribution to the likelihood of horizontal transfer over 0.025 (Euclidean distance between 5-mer

distributions; Fig. 4a). This corresponded, approximately, to the median nucleotide composition dissimilarity between random hosts from different bacterial classes. When comparing the observed and the randomized transfers, the difference in genetic dissimilarity was especially large at the class and phylum levels ($p = 1.33 \times 10^{-27}$ and $p = 3.41 \times 10^{-18}$, respectively; one-sided Wilcoxon's rank sum test), but was also significant at the order level ($p = 9.62 \times 10^{-5}$; Fig. 4a). The gene-genome incompatibility showed a similar pattern, with increasing nucleotide composition dissimilarity corresponding to decreasing likelihood of horizontal transfer (Fig. 4b). Here, however, the curve was not as steep, with negative contributions starting at values corresponding roughly to the nucleotide composition dissimilarity between a random genome and an ARG originating from a different phylum. Interestingly, the influence of gene-genome genetic incompatibility was especially pronounced for tetracycline efflux pumps (Supplementary Fig. 7), which also had the largest differences in gene-genome dissimilarity between the observed and the randomized transfers.

Our results demonstrate that genetic incompatibility—both between host genomes and between ARGs and genomes—negatively affects the transfer of resistance genes, especially between evolutionary distant hosts. To further explore how differences in genetic incompatibility may shape the spread of ARGs between bacterial phyla, we visualized the nucleotide composition of all 7609 ARG-carrying species identified in this study (Fig. 4c). Most hosts belonging to different phyla had, as expected, highly dissimilar nucleotide

compositions, suggesting high incompatibility (e.g. AT-rich Bacillota and the GC-rich Actinomycetota and, to a lesser extent, Actinomycetota and Pseudomonadota). Interestingly, however, there were hosts from different phyla that demonstrated low nucleotide composition dissimilarity—in several cases, these were more similar than many hosts within a phylum. This included, for example, Bacillota and Bacteriodota; Bacillota and Campylobacterota; and parts of Pseudomonadota and both Bacillota and Bacteriodota, suggesting that high genetic incompatibility may not necessarily prevent horizontal transfer of ARGs between evolutionarily distant hosts. However, in these cases, transfers may be limited by other factors, such as host ranges of the mobile genetic elements carrying the mobile ARGs.

### Co-occurrence in human and wastewater microbiomes increases the transfer of antibiotic resistance genes

Network analysis was used to further explore the connection between the horizontal transfer of ARGs and the co-occurrence of hosts in bacterial communities (Supplementary Fig 8). The co-occurrence of each host-pair involved in the horizontal transfer of ARGs was estimated as the proportion of metagenomic samples ($n = 20,816$) in which both hosts were present (see "Methods"). Among the frequently observed host pairs (≥5 transfers), as many as 63.3% co-occurred in at least one type of environment, while the remaining 36.7% of the host pairs were either not present or below the detection limit. Interestingly, the co-occurrence patterns were highly environment-specific. The highest diversity of co-occurring host pairs was found in the human microbiome (Fig. 5a), which encompassed 10 classes from 5 phyla. This included multiple highly connected pathogens from Pseudomonadota (e.g. *Escherichia coli*, *Acinetobacter baumannii*, *Salmonella enterica*), Bacillota (e.g. *Staphylococcus aureus*, *Clostridioides difficile*), and Campylobacterota (*Campylobacter jejuni*). Interestingly, when human microbiome samples were further stratified into major subclasses, we noticed that gut and skin had an especially high influence on the likelihood of horizontal ARG transfer, and demonstrated specific transfer patterns (macrolide and tetracycline ARGs in gut samples and aminoglycoside and beta-lactam resistance genes in skin samples, Supplementary Fig. 9). Compared to the human microbiome, the proportion of wastewater samples where host-pairs co-occurred was, overall, higher (Fig. 5, edge thickness) but taxonomically more restricted (3 phyla, 5 classes, Fig. 5b). Indeed, 23.8% of the hosts involved in horizontal transfer with measurable co-occurrence in the human microbiome were not simultaneously present in the wastewater metagenomes. In contrast, only 3.3% of hosts involved in horizontal transfer with measurable co-occurrence in the wastewater microbiome were not simultaneously present in the human microbiomes. In wastewater, connections were primarily observed between host pairs from Gammaproteobacteria (including *Acinetobacter*, *Pseudomonas*, and *Aeromonas*) and select Bacillota (including *Streptococcus* and *C. difficile*). Many pathogens from Bacilli (e.g. *S. aureus*, *Streptococcus pyogenes*) and Campylobacterota (*C. jejuni*), were, however, not found to be commonly co-occurring in wastewater, at least not at detectable levels. This is in line with previous studies, which have shown that HGT in wastewater microbial communities is most prevalent within Pseudomonadota[28]. The co-occurrence in animal and environmental bacterial communities was generally lower than in human and wastewater. The co-occurrence patterns observed in the animal microbiomes showed similarities to the human microbiome in terms of taxonomic representation, particularly concerning Pseudomonadota, Bacillota, and Campylobacterota (Supplementary Fig. 10a). The connectivity in both soil and water samples instead had an emphasis on host-pairs from Gammaproteobacteria, but with few detectable co-occurrences between other taxa (Supplementary Fig 10b, c).

Several species that have been suggested as likely recent origins of various ARGs[10] were found to frequently participate in horizontal

ARG transfer: *Klebsiella pneumoniae*, *A. baumanii*, *Acinetobacter radioresistens*, *Aeromonas caviae*, *Aeromonas media*, *Citrobacter freundii*, *Enterobacter cloacae*, *Enterobacter asburiae*, *Morganella morganii*, *Shewanella xiamenensis*, *Shewanella algae*, and *Leclercia adecarboxylata*[10]. When investigating the co-occurrence between these origin species and their frequently observed transfer partners, we found the highest number of connections in the human microbiome (108 connections, Supplementary Fig. 11), followed by wastewater (81 connections) where, again, the observed co-occurrence was generally higher. By comparison, fewer connections were identified in the animal (46), soil (24), and water (23) co-occurrence networks.

## Discussion

In this study, we analyzed the horizontal transfer of antibiotic resistance genes between evolutionarily distant bacterial hosts. By integrating data from -1 million bacterial genomes with co-occurrence patterns from >20,000 metagenomic samples, we developed machine learning models that were able to reliably predict which hosts are most likely to transfer ARGs horizontally. Our results show that genomes with similar nucleotide composition have a higher likelihood of sharing mobile ARGs, while high genetic incompatibility—at the levels generally associated with at least a class-level taxonomic difference—limits this sharing. We also demonstrate that high co-occurrence is linked to an increased likelihood of horizontal ARG transfer, however, these patterns were linked to specific environments, primarily human and wastewater microbiomes. Our study, thus, provides means to forecast the horizontal transfer of ARGs between bacterial hosts and reveals insights into how genetic and ecological factors govern the spread of resistance genes.

Our results show that genetic incompatibility has a major impact on the horizontal dissemination of ARGs. Indeed, bacterial genomes have distinct codon compositions that reflect the structure of their tRNA pool. Genes that adhere to the codon preference can, thus, be more efficiently transcribed and translated[29]. Acquired genes with suboptimal codon compositions may, consequently, come with significant fitness costs[30], and ARGs with high fitness costs will be lost over time unless there are sufficient antibiotic selection pressures[31,32]. However, translational demand may, in some scenarios, have a limited impact on the fitness costs induced by plasmids, as previously demonstrated[33]. The biochemical function of acquired ARGs needs also to be integrated effectively into its new cellular context and induce a sufficiently strong phenotype. All these evolutionary processes result in adaptations that are reflected in the nucleotide sequences[34,35]. Even so, our results consistently showed that low gene-genome nucleotide composition dissimilarity is central for facilitating horizontal transfer, both in the general model and in every resistance mechanism-specific model, except for the class B beta-lactamases (Fig. 3b, Supplementary Fig. 5). Interestingly, the importance of gene-genome nucleotide composition was especially pronounced for ARGs encoding tetracycline efflux pumps. Efflux pumps have complex structures and require interaction with several parts of the cellular machinery to be properly translocated and inserted into the cell membrane[36]. Our observations are in line with Porse et al., that demonstrated that acquired resistance mechanisms which depend on more complex interactions with the host cell are less likely to function properly in an *E. coli* host than those that interact directly with the antibiotic, such as beta-lactamases or aminoglycoside-modifying enzymes[37].

The genome-genome genetic incompatibility, similar to the gene-genome genetic incompatibility, had a major influence on the horizontal transfer of ARGs. Successful acquisition of genes often includes the incorporation of new genetic material into the recipient's genome, typically through homologous recombination guided by base-pair interactions between the two DNA molecules. It is, therefore, plausible that successful integration of DNA is more likely between bacterial

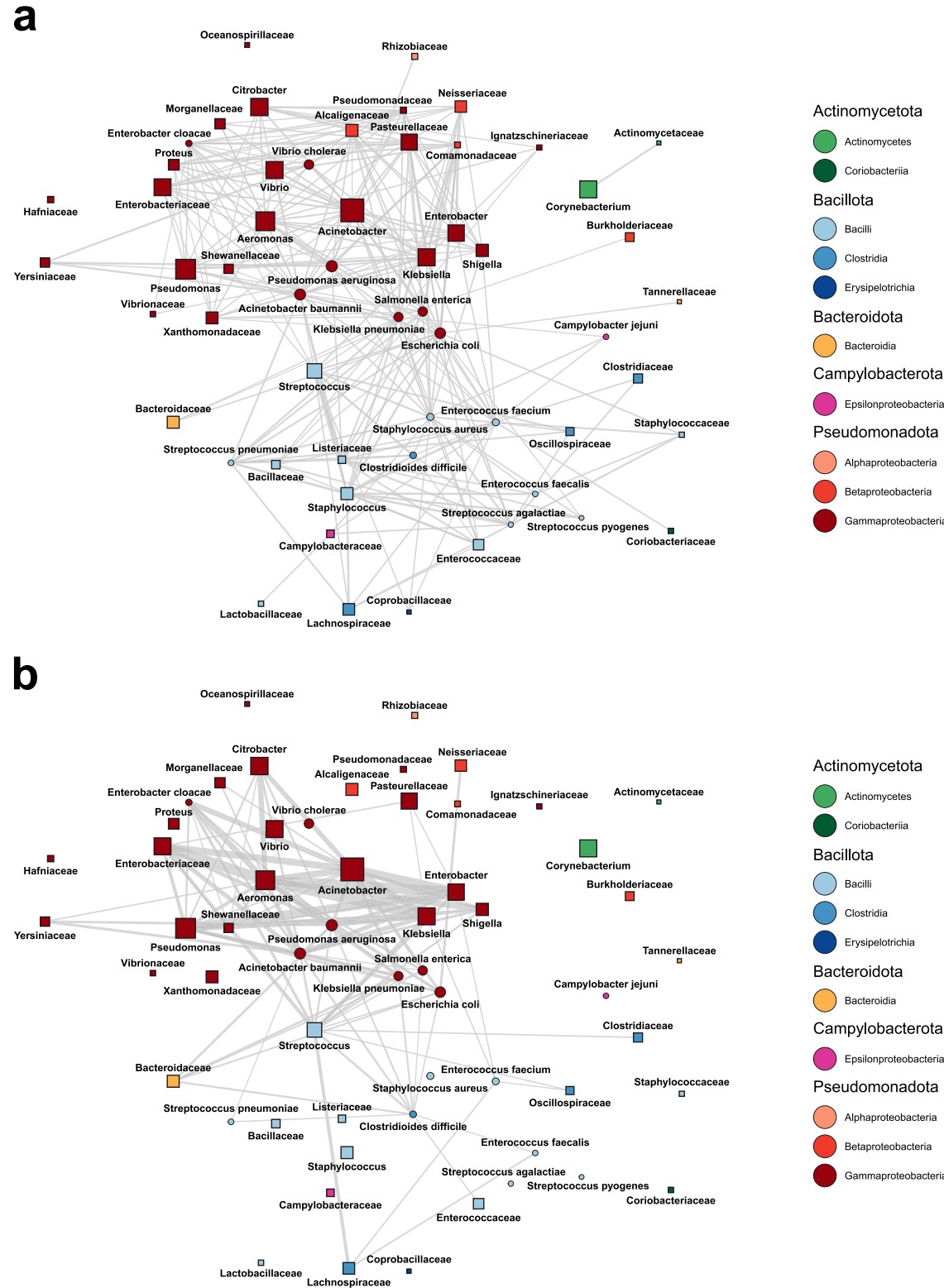

**Fig. 5 | Co-occurrence of promiscuous bacterial taxa in human and wastewater microbiomes.** In the networks, each node represents a taxon on either species, genus, or family level, which was aggregated such that individual nodes on a lower level are not part of the corresponding higher-level node(s). For each node, the size is proportional to the total number of inferred interactions associated with the taxon, and the shape indicates if the node represents a species (circle) or a higher-level taxon (square). Edges are drawn between taxa with at least an order-level distance, between which horizontal transfer was observed at least 5 times. Edge thickness indicates the maximal estimated co-occurrence of two species included in each taxon in (**a**) human samples (*n* = 3220) and (**b**) wastewater samples (*n* = 1185). If co-occurrence was measurable in less than 1% of the corresponding samples, no edge is drawn. Source data are provided as a Source Data file.

hosts with genomes that have similar nucleotide composition—a hypothesis that is supported by previous findings[14,38]. Furthermore, the transfer of ARGs is often mediated by specific biochemical functions, especially transposons, integrons, conjugative elements, and DNA replication systems, and the presence of these elements in both donor and recipient genomes will increase the similarity of their nucleotide compositions[20,39].

Our findings, thus, show that genetic incompatibility is one of the most influential factors for successful horizontal ARG transfer. It should, however, be emphasized that this barrier reflects multiple genetic signals—some of which do not necessarily correspond to the evolutionary distance between bacteria. In fact, the nucleotide composition of bacteria can, in some situations, be more similar between genomes from different phyla than between lower taxonomic levels (Fig. 4c). Indeed, inter-phyla transfers were consistently associated with much lower genetic incompatibility than expected based on the evolutionary distance between the hosts (Fig. 4a, b). Thus, we argue that genetic incompatibility acts as a barrier and shapes how ARGs are disseminated between bacteria, including acquisition by and circulation among pathogens.

High ecological interaction facilitates the transfer of genetic material[22,40]. This is reflected in our results, where bacterial co-occurrence in metagenomes was associated with an increased likelihood of horizontal ARG transfer. In particular, the human microbiome and wastewater bacterial communities showed the highest co-occurrence for bacterial hosts involved in horizontal ARG transfer. By contrast, there were fewer co-occurring hosts in the animal microbiomes and the external environments (Supplementary Fig. 10). The human gut and wastewater are both environments where antibiotics may be present and provide the necessary selection pressures to promote ARG proliferation[41,42]. Indeed, several shotgun metagenomic studies have shown that these environments typically contain a high abundance of ARGs together with mobile genetic elements that can facilitate horizontal dissemination[43–45]. Our results, thus, reinforce these findings, but also suggest that the large majority of ARG transfers that are currently documented in sequence repositories—in which clinical isolates are abundant—have predominately happened in the human and wastewater microbiomes.

Many of the co-occurring hosts were specific to a single or a few environments. Indeed, several ARGs, such as APHs and class A, C, and D beta-lactamases, were predominantly transferred between bacteria co-occurring in the human-associated bacterial communities, while others, e.g. AACs, were more frequently transferred by bacteria co-occurring in wastewater (Fig. 3c). In fact, almost one fourth (23.8%) of the hosts that frequently transferred ARGs and co-occurred in the human microbiome, were not simultaneously detected in a single wastewater metagenome. Vice versa, only 3.3% of the hosts that frequently transferred ARGs and co-occurred in wastewater metagenomes were not present in a single human metagenome. The level of co-occurrence, i.e. the proportion of samples in which both hosts were simultaneously detected, was, however, generally higher in wastewater compared to human microbiomes, though it should be noted that this is influenced by the number of samples from each category as well as their inherent level of heterogeneity. Indeed, previous studies have hypothesized that distinct HGT networks are associated with the gut microbiomes of different individuals[46], which could further explain the lower generalizability of the human samples. This trend could be seen for many promiscuous bacteria, including hosts that were identified as recent origins of mobile ARGs by Ebmeyer et al. 2021[10]: e.g. *M. morganii* and *S. algae* co-occurred with their respective identified transfer partners at a detectable level in the human microbiome, but not in wastewater. Our results, thus, suggest that the transfers between some bacterial hosts are limited to—or are at least more likely in—certain environments. Thus, no single environment can be regarded as the only facilitator—or the sole 'hotspot'—for the horizontal transfer of

antibiotic resistance genes. Indeed, the transfer of ARGs will, to a large extent, be governed by the taxonomic composition of the microbial community, which varies substantially between environments, including the human and wastewater microbiomes[47,48]. However, previous research has shown that bacteria from disparate environmental origins can coalesce and migrate between environments in microbiotic particles[49], which further suggests a way for ARGs originating in external environments to interact with host-associated microbiomes. Our results, thus, show the importance of a one-health perspective[50,51] on how ARGs transfer between bacterial hosts and underscore the need for management and surveillance strategies that are not limited to specific environments.

Roughly half (53.6%) of all species-pairs associated with horizontal ARG transfer were estimated to co-occur in at least one type of environment. Accordingly, the remaining half consisted of host pairs that could rarely (<1% of samples, see Methods) or never be detected in any of the more than 20,000 included metagenomic samples (Supplementary Fig. 8). This reflects species with an abundance below the detection limit, but only to some extent, as 54.3% of the missing host-pairs consisted of species that both could be detected individually. This suggests these pairs either co-occur in less well-studied bacterial communities or that the transfers often include one or more intermediary hosts that are not represented in the current sequence repositories, where environmental species are significantly under-represented. In fact, even if our analysis was based on almost one million bacterial genomes, only ~6000 successful horizontal ARG transfers could be identified—a clear underestimation. Consequently, some mechanism-specific models (e.g. the Mph, Qnr, and Tet enzyme mechanisms) could not be reliably trained and evaluated due to a lack of observed transfers (see Methods). Nevertheless, our results demonstrate several significant associations—both for the general and the mechanism-specific models. This indicates that the data is still sufficiently rich to infer several of the factors that govern the successful horizontal transfer of ARGs. Furthermore, the low co-occurrence observed in water and soil suggests that these environments are less likely to spread clinically relevant ARGs between known bacterial hosts—although this may not be true for yet-to-be-discovered resistance determinants and the many uncultured species present in these environments[52]. Our results should, therefore, be considered conservative, and additional factors influencing the successful horizontal transfer of ARGs will likely be discovered as the genomic and metagenomic repositories become more comprehensive. Finally, to circumvent the lack of comprehensive data on cell envelope structure we used Gram staining as a proxy, which was assigned to each genome based on the consensus of the respective phyla. There are, however, examples of taxa that deviate from this rule, in particular Negativicutes from Bacillota[53]. However, less than 3% of the transfers were associated with Negativicutes, and considering the relatively low influence of this factor, the impact on the overall performance is limited. However, it is plausible that our model has a reduced performance when predicting the horizontal transfer of ARGs for such taxa.

Pathogens become resistant to antibiotics by acquiring ARGs transferred from evolutionary distant bacterial hosts. This ongoing evolutionary process remains elusive, which makes the implementation of countermeasures limiting the flow of ARGs challenging. Our results demonstrate that data-driven approaches combined with machine learning enable accurate analysis of how ARGs are transferred between bacterial hosts. Particularly, our results emphasize that differences in genomic composition and co-occurrence of hosts in bacterial communities are two key factors that shape the dissemination of ARGs. We conclude that genetic incompatibility and ecological connectivity significantly impact the evolutionary processes leading to the proliferation of antibiotic resistance among environmental, commensal, and disease-causing bacteria. We also conclude that predictive models can play an important role in detecting emerging resistance

determinants and in assessing the risk that they will become established in human pathogens.

## Methods

### Identification of horizontally spread antibiotic resistance genes

A total of 1,150,159 bacterial genomes were downloaded from the NCBI Assembly (2022-04-04)[54]. To ensure the robustness of our analysis, 282,841 genomes that did not pass NCBI's taxonomy check and/or where contamination was suspected based on the provided annotations, were removed from consideration. After filtration, the database still included some assemblies derived from metagenomes, however, these only encompassed 6449 genomes (0.74%). Next, fARGene v0.1[55] was used to screen the remaining 867,318 genomes (Supplementary Data 1) for ARGs. fARGene is a software that can identify both known and uncharacterized resistance genes using a total of 22 hidden Markov models, each specifically optimized to detect a gene class where all members share an evolutionary history. Among the included gene classes, six encode aminoglycoside acetyltransferases (AAC(2′), AAC(3) class 1, AAC(3) class 2, AAC(6′) class 1, AAC(6′) class 2, AAC(6′) class 3), three encode aminoglycoside phosphotransferases (APH(2″), APH(3′), APH(6′)), six encode beta-lactamases (Class A, Class B1/B2, Class B3, Class C, Class D1, Class D2), two encode Erm 23S rRNA-methyltransferases (Erm type A, Erm type F), and the remaining five classes encode Mph macrolide 2′-phosphotransferases (Mph), quinolone resistance genes (Qnr), tetracycline efflux pumps (Tet efflux), tetracycline inactivating enzymes (Tet enzyme), and tetracycline ribosomal protection genes (Tet RPG). Here, we screened the downloaded genomes for all 22 gene classes, and each class was analyzed separately. For each gene class, the predicted protein sequences were clustered at 100% amino acid identity using USEARCH v8.01445[56] with parameters '-cluster_fast -id 1' to remove redundant sequences. A multiple sequence alignment was then created using mafft v7.458[57] with default parameters, and a phylogenetic gene tree was reconstructed from the alignment using FastTree v2.1.10[58] using default parameters.

Each of the resulting 22 gene trees was searched for instances where ARGs showed evidence of having spread successfully between taxonomic orders. Horizontally transferred ARGs were detected by traversing the phylogenetic trees, from the leaves towards the root, searching for nodes where the descendant leaves represented genes identified in hosts from different taxonomic orders (at least one host genome from each order). Nodes where horizontal transfer had previously been identified among its descendants, were excluded, thereby ensuring that (1) each gene found to have spread horizontally was counted only once and (2) further analysis would be based only on the most recent transfer events. For leaves where genomes from multiple orders were found to encode identical resistance protein sequences, all possible combinations of orders were included in the downstream analysis. The phylogenetic trees and corresponding observations of horizontal transfer for each of the 22 fARGene gene classes were visualized using the ggtree R package v2.0.0[59]. For each leaf in the trees, the amino acid identity of its closest known homolog was calculated using BLASTp from BLAST+ v2.10.1[60] using CARD as a reference database (downloaded 2023-03-28)[61] and visualized on the perimeter of the tree together with the host phylum (Supplementary Fig. 1). To assess the recency of the transfer event associated with each observation, the protein sequences carried by the two host taxa were aligned against each other using BLASTp from BLAST+ v2.10.1[60], and the highest observed sequence identity was recorded.

### Analysis of genetic incompatibility between genes and genomes

An identified transfer comprised two sets of genomes from NCBI Assembly—where all genomes in a set belonged to a single taxonomic order, different from the second set—and the ARGs carried by these genomes. For each observed instance of horizontal ARG transfer, the genetic incompatibility of the resistance genes and genomes involved was evaluated. Using genome and ARG sequences, we calculated two estimates: (1) the genome-genome incompatibility measured by the nucleotide composition dissimilarity of the two genome sets, and (2) the gene-genome incompatibility measured by the nucleotide composition dissimilarity of the transferred ARG and its host genome(s). For both incompatibility measures, the genome sequences and their ARGs were divided into 5-mers, and their respective 5-mer distributions were calculated. The genome-genome incompatibility was quantified as the Euclidean distance between the mean 5-mer distributions of the two genome sets. The gene-genome incompatibility was estimated in the following way. A single ARG sequence was randomly selected from the two sets, and the Euclidean distance between the selected ARG's 5-mer distribution and the mean 5-mer distributions for each genome set was calculated. The gene-genome incompatibility was quantified as the maximum of these two distances since it was inferred as the highest dissimilarity where the gene is still expected to function. The smaller of the two distances was discarded. Additionally, the proportional difference in mean genome size between the genome sets involved in each observed transfer was calculated.

### Estimation of bacterial co-occurrence from metagenomic data

To measure the ecological connectivity of the observed host bacteria in different environments, we downloaded a total of 24,417 metagenomic samples from the Earth Microbiome Project (2022-02-10)[62] and the Global Water Microbiome Consortium[63]. First, the nucleotide sequences of all operational taxonomic units (OTUs) present in the metagenomic dataset were aligned against all genomes available in NCBI Assembly using BLAST+ v2.10.1 with default parameters[60]. For each genome, the best matching OTU based on sequence identity was identified requiring a coverage >90% and sequence identity >97%. The matches were then used to assign each genome carrying horizontally disseminated ARGs to an OTU, where multiple genomes were allowed to map to the same OTU. Based on these strict criteria, only 293,295 of the 867,318 included genomes (33.82%) could be assigned an OTU (Supplementary Data 2). Since random forest models are unable to handle missing values, this further resulted in the removal of 3409 of the total 6276 identified transfers (54.32%).

Next, the co-occurrence of the representative OTUs was estimated in metagenomic samples from five different environment categories (divided based on the available metadata) which were balanced regarding the number of samples: animal (4352 samples), human (3244 samples), soil (4137 samples), water (7898 samples), and wastewater (1187 samples). Here, four samples with insufficient sequencing depth (<10,000 total reads) were removed before analysis, as were 3597 samples that did not correspond to any of the aforementioned environment categories, leaving a total of 20,816 samples. For each observed transfer, the co-occurrence in a given environment was estimated by first considering all possible pairs of OTUs represented by host genomes from different orders. Next, for each of these pairs, we calculated the proportion of metagenomic samples from that environment where both OTUs were considered present (≥3 reads). Finally, we estimated the overall co-occurrence for that transfer and environment as the mean of the calculated proportions.

### Inference on horizontally spread ARGs using random forest classifiers

The following twelve features were used as input to train random forest models to separate between observed and randomized transfers, using the randomForest R package v 4.7-1.1 with default hyperparameters (ntree = 500, mtry = 3, nodesize = 1)[64]: the genome-genome nucleotide composition dissimilarity and the difference in genome sizes between the two genome sets associated with the horizontally spread ARGs; the maximum gene-genome nucleotide composition dissimilarity between the horizontally spread ARG and its host

genomes; the estimated co-occurrence of the hosts in animal, human, soil, water, and wastewater microbiomes; three binary features describing the Gram staining properties of the host pair based on their phyla (G+/G+, G−/G−, G+/G−); and one categorical feature denoting the gene class of the transferred ARG (categories taken from fARGene). Before training the models, 3419 observations that had a missing value in any of their input variables (54.48%) were removed, since random forest models are unable to handle missing values. In addition, instances where >10 potential transfers were observed for genes encoding identical proteins were downsampled such that only 10 randomly selected host pairs were retained. This was done to avoid bias in the input data, which would otherwise be dominated by a small number of very well-spread ARGs.

In addition to the positive dataset encompassing the observed transfers, a negative dataset was generated from the large set of predicted ARGs. Briefly, negative observations were generated by randomly sampling two leaves from the phylogenetic gene trees that represented protein sequences encoded by hosts with at least an order level difference in their taxonomy, without replacement until no more leaf pairs could be generated. For leaves representing 100% identical protein sequences encoded by multiple bacterial orders, a single order was randomly picked. In addition, if a selected leaf represented a protein encoded by >1000 host genomes, we randomly selected 1000 genomes for further analysis to increase computational efficiency. For each resulting random pair of leaves, features were calculated identically as for the true data points. The correlations between the input variables were generally considered low for both the positive and negative datasets (Supplementary Fig. 12). Finally, when creating the models, the negative dataset was downsampled to include an equal number of negative observations for each fARGene gene class, matching the corresponding number of true observations.

In total, eight models were created, each using 70% of its input data as training data and 30% as test data, split randomly. The first random forest classifier was trained on horizontally transferred ARGs representing all analyzed classes of ARGs ($n = 1565$ after removing missing values and downsampling redundancy). The remaining seven classifiers were trained on horizontally spread ARGs associated with a specific resistance mechanism: aminoglycoside acetyltransferases (AAC, $n = 251$), aminoglycoside phosphotransferases (APH, $n = 295$), class A, C, D beta-lactamases ($n = 346$), class B beta-lactamases ($n = 104$), Erm 23S rRNA methyltransferases ($n = 128$), tetracycline efflux ($n = 154$), and tetracycline ribosomal protection genes (RPGs, $n = 168$), respectively. No model was trained exclusively on horizontally spread Mph macrolide 2′-phosphotransferases, $qnr$ quinolone resistance genes, or tetracycline inactivation enzymes since too few observations of these were detected in the original dataset (<100 observations after removing missing values).

The performance of each random forest model was evaluated based on the observed AUROC, calculated using the pROC R package v1.18[65], as well as the sensitivity and specificity calculated from the confusion matrix, after using the classifier to make predictions on test data. To account for the variability from random sampling, ten iterations of each model were generated using a resampled negative dataset, and the mean and standard deviations of each performance metric were recorded. Then, for each model, the feature importance, in terms of MeanDecreaseAccuracy at a 0.5 cut-off, and corresponding $p$-value were calculated using the rfPermute R package v2.5.1[26]. The sign of the MeanDecreaseAccuracy – representing whether an increased value of a feature was generally indicative of horizontally spread ARGs (+) or not (-) – was calculated using the rfUtilities R package v2.1-5[66]. Partial dependence plots were generated for the features representing genetic incompatibility using the pdp R package v0.8.1[67]. To analyze the general nucleotide composition dissimilarity of different taxa, multi-dimensional scaling was performed on the 5mer distributions from 7,609 selected genomes of the highest available

quality representing all unique species from the major phyla (Actinomycetota, Bacteroidota, Bacillota, Campylobacterota, Pseudomonadota) found to carry an ARG using the 'metaMDS' function from the vegan R package v2.6-4[68], with distance set to 'euclidean' (Fig. 4c).

Since the environment categories that we chose for the co-occurrence variables were quite broad (with the exception of the wastewater category)—and therefore contained an inherent heterogeneity with regard to taxonomic composition—we also wanted to investigate whether the use of more specific co-occurrence variables would improve the models. To test this, we divided the human samples into human gut ($n = 714$), human skin ($n = 1456$), and human oral ($n = 702$), while the water samples were divided into freshwater ($n = 5282$) and marine ($n = 1259$). The human and water samples that did not fit into any of the smaller categories were discarded ($n = 348$, $n = 1347$, respectively). The updated models were created and analyzed based on feature importance as described above (Supplementary Fig. 9). Ultimately, however, these models showed negligible differences in performance compared to the previous models (Supplementary Fig. 13). This, in combination with the large number of metagenomic samples that had to be discarded led us to not pursue these environment-specific models further.

### Network analysis of species frequently associated with horizontal spread of ARGs

Using the full set of ARGs observed to have been horizontally transferred between different bacterial orders, we calculated the number of transfers in which individual species with an order-level taxonomic difference were observed together in different genome sets. Many of the transfers involved multiple species in each genome set, and here, we generated all possible combinations of species from the first and the second genome set, inferring all of these as potential transfer routes for the ARG. For each species pair, the co-occurrence of the representative OTUs was then calculated as previously described. Next, we removed all species pairs that were observed <5 times in total, keeping only the pairs with strong connections. To reduce the complexity and enable an easier overview, we opted to aggregate the included species at higher taxonomic levels based on their overall importance. Thus, aside from a selection of important pathogens, the species in the remaining list were aggregated first to the genus level, and if a genus was not sufficiently connected to other taxa (<2500 observed transfers) it was further aggregated to the family level. From the aggregated list, we generated gene transfer networks using the igraph v1.4.3 R package[69], which were visualized using the GGally v2.1.2 R package[70]. Networks were generated for each of the five environment categories (human, animal, soil, water, wastewater), where the thickness of the edges represented the co-occurrence of the connected taxa in the corresponding environment (i.e. the proportion of samples from this environment in which the taxa co-occurred). No edges were drawn between taxa with measurable co-occurrence in <1% of the respective samples (Fig. 5, Supplementary Fig. 10a–c). For reference, we also included a network that only showed the observed connections between taxa, and omitted any information about co-occurrence (Supplementary Fig. 10d). To provide more detail about the co-occurrence patterns of individual species, we also generated a set of equivalent but more complex networks by using the same methodology but not aggregating taxa (Supplementary Fig. 8).

Finally, we generated a third set of networks describing the transfer and co-occurrence patterns of species that have been proposed as recent origins of mobile ARGs[10]. These networks were generated in largely the same way, with a few exceptions. First, all species were aggregated to higher taxonomic levels except for the selected origin species. Then, no edges were drawn between nodes that did not

represent the origin species, while observed transfers between an origin species and other taxa that lacked measurable co-occurrence above the cut-off were drawn as dashed lines (Supplementary Fig. 11).

**Reporting summary**

Further information on research design is available in the Nature Portfolio Reporting Summary linked to this article.

## Data availability

All raw data used for this study have been retrieved from public repositories including NCBI[54], the Earth Microbiome Project (https://earthmicrobiome.org/)[62], and the Global Water Microbiome Consortium (http://gwmc.ou.edu/)[63]. Accession numbers of the analyzed genomes and metagenomes are provided in Supplementary Data 1. The intermediary data generated in this study are available via Zenodo at https://doi.org/10.5281/zenodo.14901409[71]. Source data are provided with this paper.

## Code availability

Scripts used to generate the positive and negative datasets used to train the random forest models are available via GitHub at https://github.com/davidgllund/factors_influencing_HGT_of_ARGs[72] and Zenodo at https://doi.org/10.5281/zenodo.13827913[73]. Scripts and files used to analyze the datasets and produce all the main and Supplementary Figs. from the paper are available via Zenodo at https://doi.org/10.5281/zenodo.14901409[71].

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

## Acknowledgements

This research was supported by the Swedish Research Council (VR) (2018–02835, 2018-05771,2019–03482 and 2022-00945). Funding sources took no part in the design, analysis, or interpretation of the results.

## Author contributions

D.L., M.P.-M., J.I.-D., S.E., D.G.J.L., A.J., and E.K. designed the study and developed the methodology. D.L. and M.P.-M. collected the data and implemented the computational analysis for the identification of horizontally transferred resistance genes. D.L. implemented additional computational analysis including computation of relevant factors and training of random forest models. D.L., J.I.-D., S.E., D.G.J.L., A.J., and E.K. discussed the results and their implications. D.L. and E.K. drafted the manuscript. All authors edited and approved the final manuscript.

## Funding

## Competing interests

The authors declare no competing interests.
