## [Transparent Peer Review file · Nature Communications]

Genetic compatibility and ecological connectivity drive the dissemination of antibiotic resistance genes

Corresponding Author: Dr Erik Kristiansson

Version 0:

Reviewer comments:

Reviewer #1

(Remarks to the Author)

This manuscript focuses on the critical importance of understanding the factors influencing the spread of antibiotic resistance genes (ARGs) through horizontal gene transfer (HGT), which is essential for developing strategies to manage the antibiotic resistance crisis. The authors conducted a comprehensive study, including the analysis of approximately 1 million bacterial genomes and over 20,000 metagenomes. They developed random forest models to predict horizontal ARG transfer between bacteria.

The study found that genetic incompatibility, measured as nucleotide composition dissimilarity, negatively influences the likelihood of ARG transfer between evolutionarily divergent bacteria. In contrast, environmental co-occurrence increased the likelihood of ARG transfer, especially in human and wastewater environments. The study identified 6,276 horizontal transfers of ARGs, with aminoglycoside phosphotransferases (APHs) and class A, C, or D beta-lactamases being the most common. The random forest models demonstrated high accuracy in predicting horizontal ARG transfer.

These findings provide new methods for predicting the spread of ARGs and offer insights into the mechanisms governing this evolutionary process. This research could aid in developing countermeasures to limit the spread of new ARGs and preserve the efficacy of current and future antibiotics. The study concludes that genetic incompatibility and ecological connectivity significantly impact the horizontal transfer of ARGs. Predictive models play a crucial role in detecting emerging resistance determinants and assessing the risk of their establishment in human pathogens.

Overall, the manuscript provides valuable information but could benefit from the following minor corrections:

- Adding more details about the random forest models, such as the specific parameters used and the rationale behind their selection, would enhance reproducibility and understanding.
- The results section is comprehensive however, providing more context for the findings, like comparing them with previous studies or discussing their implications in greater detail, might be useful.

(Remarks on code availability)

NA

Reviewer #2

(Remarks to the Author)

Lund et al. present a very timely and important manuscript assessing the contributing factors of dissemination of ARGs. As they outline in their Introduction, it is pressing that we understand this issue with implications in healthcare, food production pipelines, and for many other sectors that currently heavily rely on antimicrobials.

Overall, I think the work is well done; however, in places lack of details make it hard to assess. In particular, I didn't understand how ARGs were clustered (section line 362). If I am understanding correctly, multiple ARG homologs were combined into a single gene model based on the antimicrobial that they confer resistance to then used to construct gene trees. If this is the case, I disagree with this methodology and would like to see justification as to why multiple homologs were combined into a "gene" tree in this way. If this is not the case, the text needs to be improved to help the reader understand the methodology.

Indeed, throughout the manuscript, improvements to the text are necessary to improve understandability. I also include methodological questions. Comments in order of appearance:

Line 73 – Typo, 2.6 million

Line 80 – The number of metagenomes associated with some microbiomes (e.g. human and wastewater) are likely to outnumber many other types of communities by the nature of the types of metagenomic sequencing that we tend to do. This should be included as a caveat of this work. I do see it briefly mentioned at line 312 but a more thorough discussion/investigation is needed.

Figure 2 – Acronyms used in the Figure need to be included in the legend (e.g., ARG, Resistance mechanisms). A brief summary of the resistance mechanisms should be included in the text (Introduction or appropriate Results section) to put the Results in context. (And in all Figures/Sup Figs).

Figure 2d – y axis is missing a title.

Line 161/Fig4ab- I don't understand what I'm looking at here, and I suspect most of your desired readership won't either. A few sentences explaining partial dependence analysis and walking your audience through your analysis is needed here. Why were no other factors looked at in this way? Only genome-gene 5mr and mean 5mr.

Sup Fig 9 missing?

Line 212-3 – Was this quantified? “Compared to the human microbiome, the co-occurrence of host-pairs in wastewater samples was, overall, higher.”

Sup Fig 10d- Is this across all samples?

Sup Fig 11- Legend does not match the mention of this Figure in the text. What is being represented? How does this differ from Sup Fig 10?

Line 308- Is this new data being introduced in the Discussion?

Line beginning 314- I don't understand how the example provided (*M. morgani* and *S. algae*) corresponds to the point of recent origins of ARGs?

Line 355- Typo, ARGs.

Line 363- How many genomes were downloaded? How many genomes were there before and after curation? How was “where contamination was suspected” determined?

Line 370 – Typo, ARG.

Line 370 – “predicted translated ARGs were clustered at 100% amino acid identity” what does this mean here? To me, that reads as individual ARG homologs were determined from the results of fARGene by clustering for similarity at 100%; however, that doesn't really make sense with the next steps of MSA alignment, tree building. How were individual ARG gene clusters identified here?

Line 374- What is the distribution of identified ARG gene clusters/individual protein types? Does Sup Fig 1a, for eg, include a single ARG homologous gene cluster? If not, how many homologs per phylogeny and what justifies including them in a single gene tree?

Line 414- Have the authors considered the implications of focussing on more environmental-type datasets? For e.g. given the focus on human microbiomes in the text, I am surprised that metagenomic data from the Human Microbiome Project was not included?

Line 419- How were cases where an OTU matched multiple genomes of different taxonomic assignments dealt with? OTUs are short and often not specific to a given genome, strain, or even species. Why wasn't a widely used approach for OTU taxonomic calling used here?

Line 426- How was co-occurrence determined? No thresholds or analyses tools/methods mentioned.

Line 446- How many were removed?

Figure 2 – Has any correction been made for more commonly sequenced organisms? For e.g. if a HGT from *E.coli* to *Staph aureus* was identified 100x but a HGT between 2 less-well studied organisms was identified only 10x, was some sort of normalization performed?

Sup Fig 1- Threshold for the number of times a HGT was seen to call it a HGT? For e.g. if the ARG is identified in a phyla 100 times and in a second phyla once, will that be counted?

Fig 3a- The Results section is missing an explanation of which factors were taken into account in the ML RF model: what they are, how they were calculated, why they were included over other factors? For example, the sentence starting at line 143 and pointing to Fig 3b discusses genome/gene differences in nucleotide composition; is this the genome-gene 5mer distance data in Fig 3b? I appreciate that this is included in the Methods but, as written, this isn't seen to the reader until after the Results are presented- some of this could be rectified by changing the axis labels.

Line 411- First mention of genome size in the text; would be helpful to bring this into the Results section.

Discussion of the impact of fragmented MAGs might have on this work?

In the Introduction, the authors outline how genes are HGTed including that many genes will exist on extra chromosomal elements. What was the distribution of ARGs on extra-chromosomal elements vs. chromosomes? In the metagenomic data, how were extra-chromosomal elements linked to chromosomes (and organisms)?

(Remarks on code availability)

Reviewer #3

(Remarks to the Author)

This interesting work is based on the application of a machine learning strategy leveraging over 2.6 million ARGs identified in almost 1 million genomes and predicting horizontal gene transfer of antibiotic resistance genes (ARGs). The work, using an extensive metagenomic analysis, also explores ecological connectivity and the phylogenetic neighborhood and consequently predicts horizontal ARG transfer between bacterial hosts. The results of such huge research do not provide really novel results or suggestions, but certainly strongly reinforce previous hypotheses and findings. A weakness of the manuscript is the inherent bias of databases, with an anthropocentric composition of the analyzed microorganisms, preventing a generalization of the results to the microbiosphere. However, the data provided are certainly of interest for future research in the field, and in my opinion, the manuscript deserves publication, with some additions/discussions that I am mentioning below.

Line 52. This referee agrees that the transmission of genes from environmental/commensal organisms to pathogens is a critical question in AMR research.

Line 58: the authors, after mentioning transformation, should add the "new mechanism" of AMR transfer by extracellular vesicles. See, for instance, Qu S, Zhang Y, Weng L, Shan X, Cheng P, Li Q, Li L. The role of bacterial extracellular vesicles in promoting antibiotic resistance. *Crit Rev Microbiol.* 2024 Nov 4:1-18. doi: 10.1080/1040841X.2024.2423159

Line 67. This is a hot point in our days. Add the recent reference: Akob DM, Oates AE, Girguis PR, Badgley BD, Cooper VS, Poretsky RS, Tierney BT, Litchman E, Whitaker RJ, Whiteson KL, Metcalf CJE; Ecology Evolutionary and Biodiversity Retreat Participants. Perspectives on the future of ecology, evolution, and biodiversity from the Council on Microbial Sciences of the American Society for Microbiology. *mSphere.* 2024 Nov 21;9(11):e0030724. doi: 10.1128/msphere.00307-24.

Line 81. The authors should acknowledge that the available repositories are predominantly populated with pathogenic, clinical, and sewage isolates, thereby introducing a bias that should be addressed for the sake of objectivity. The most viable solution is to encourage future research to quantify (or at least estimate) the proportion of different microorganisms in the various ecological compartments/patches and normalize the results accordingly.

Line 147. That is an oversimplification. Gram staining does not mean a different cell wall (muropeptide) composition, in any case, a different "cell envelope". Note that many Gram-negative groups (as Negativicutes) are phylogenetically close with others that are Gram-positive by staining. It could be nice to see how these "pseudo-Gram negatives" behave with other "true" Gram negatives. In any case that should be discussed.

Line 151. The authors should make clear in their message that the key aspect of the dissemination of ARGs by horizontal transfer depends on the promiscuity of the mobile genetic elements carrying ARG genes.

Line 158. That reflects the over-representation of human and animal strains in databases, and not necessarily the reality in nature. See my previous comment.

Line 182. This point cannot be presented here as a "discovery", as it was suggested decades ago in the microbiological literature (part of it can be found for instance for sure in in your reference 3)

Line 192. Again, that should be considered in light of the promiscuity of mobile genetic elements.

Line 225. This referee agrees with the fact that connectivity and niche neighborhoods facilitate ARG transmission. An important point to consider (and eventually discuss in the Discussion section) is the "coalescence" or "merging of microbiomes" of different origins in microbiotic particles (Baquero, F., Coque, T. M., Guerra-Pinto, N., Galán, J. C., Jiménez-Lalana, D., Tamames, J., & Pedrós-Alió, C. (2022). The influence of coalescent microbiotic particles from water and soil on the evolution and spread of antimicrobial resistance. *Frontiers in Environmental Science*, 10, 824963.). The co-occurrence of organisms of disparate origins can be explained in such a way.

Line 247. Of course, not only ARGs are submitted to horizontal gene transfer. That is why in the work referenced (21) the study included all genes from the "accessory genome" (ARGs are part of this genome). Mobile genetic elements able to transfer non-ARG traits can be exploited to disseminate ARGs if they are able to capture (from the chromosome or from other mobile genetic elements) these antibiotic resistance genes.

Lines 254 or 267. However, translational demand is not influencing plasmid-associated fitness costs (Rodríguez-Beltrán, J., León-Sampedro, R., Ramiro-Martínez, P., de la Vega, C., Baquero, F., Levin, B. R., & San Millán, Á. (2022). Translational demand is not a major source of plasmid-associated fitness costs. *Philosophical Transactions of the Royal Society B*, 377(1842), 20200463).

Line 274. That is what I mentioned above.

Line 323. That is discussed in detail in Hernando-Amado, S., Coque, T. M., Baquero, F., & Martínez, J. L. (2019). Defining and combating antibiotic resistance from One Health and Global Health perspectives. *Nature microbiology*, 4(9), 1432-1442.

(Remarks on code availability)

Version 1:

Reviewer comments:

Reviewer #1

(Remarks to the Author)

I am satisfied that the revised manuscript has addressed my comments and I have no other comments or concerns.

(Remarks on code availability)

Reviewer #2

(Remarks to the Author)

The authors have addressed all comments raised.

(Remarks on code availability)

Reviewer #3

(Remarks to the Author)

I am fully satisfied with the modifications included by the authors in response to de referee's queries, and consequently I endorse the publication of the manuscript.

(Remarks on code availability)

Response to review comments

Reviewer 1

This manuscript focuses on the critical importance of understanding the factors influencing the spread of antibiotic resistance genes (ARGs) through horizontal gene transfer (HGT), which is essential for developing strategies to manage the antibiotic resistance crisis. The authors conducted a comprehensive study, including the analysis of approximately 1 million bacterial genomes and over 20,000 metagenomes. They developed random forest models to predict horizontal ARG transfer between bacteria.

The study found that genetic incompatibility, measured as nucleotide composition dissimilarity, negatively influences the likelihood of ARG transfer between evolutionarily divergent bacteria. In contrast, environmental co-occurrence increased the likelihood of ARG transfer, especially in human and wastewater environments. The study identified 6,276 horizontal transfers of ARGs, with aminoglycoside phosphotransferases (APHs) and class A, C, or D beta-lactamases being the most common. The random forest models demonstrated high accuracy in predicting horizontal ARG transfer.

These findings provide new methods for predicting the spread of ARGs and offer insights into the mechanisms governing this evolutionary process. This research could aid in developing countermeasures to limit the spread of new ARGs and preserve the efficacy of current and future antibiotics. The study concludes that genetic incompatibility and ecological connectivity significantly impact the horizontal transfer of ARGs. Predictive models play a crucial role in detecting emerging resistance determinants and assessing the risk of their establishment in human pathogens.

Overall, the manuscript provides valuable information but could benefit from the following minor corrections:

1. *Comment:* Adding more details about the random forest models, such as the specific parameters used and the rationale behind their selection, would enhance reproducibility and understanding.

Reply: We agree with the reviewer that including the specific hyperparameter values, rather than just stating that we used the default values, would improve the transparency of the manuscript. We have now included this in the Methods section (Line 649).

2. *Comment:* The results section is comprehensive however, providing more context for the findings, like comparing them with previous studies or discussing their implications in greater detail, might be useful.

Reply: Following the reviewer's suggestion, we have added additional context for our findings (at Line 308 and Line 436). This includes comparison to additional previous studies, in particular Fang et al. 2024 (doi: <https://doi.org/10.1016/j.scitotenv.2023.168908>) and Kent et al. 2020 (doi: <https://doi.org/10.1016/j.scitotenv.2023.168908>)

<https://doi.org/10.1038/s41467-020-18164-7>). Additionally, based on suggestions from Reviewer 3 we have also included discussion of our results in the contexts of Baquero et al. 2022 (doi: <https://doi.org/10.3389/fenvs.2022.824963>, Line 448) and Rodríguez-Beltran et al. 2022 (doi: <https://doi.org/10.1098/rstb.2020.0463>, Line 365).

Reviewer 2

Lund et al. present a very timely and important manuscript assessing the contributing factors of dissemination of ARGs. As they outline in their Introduction, it is pressing that we understand this issue with implications in healthcare, food production pipelines, and for many other sectors that currently heavily rely on antimicrobials.

Overall, I think the work is well done; however, in places lack of details make it hard to assess. In particular, I didn't understand how ARGs were clustered (section line 362). If I am understanding correctly, multiple ARG homologs were combined into a single gene model based on the antimicrobial that they confer resistance to then used to construct gene trees. If this is the case, I disagree with this methodology and would like to see justification as to why multiple homologs were combined into a "gene" tree in this way. If this is not the case, the text needs to be improved to help the reader understand the methodology.

Reply: We thank the reviewer for the constructive review and appreciate that this section might be a bit unclear. We believe that the reviewer has misinterpreted how the gene trees were created. The genes were not clustered based on the antibiotic to which they confer resistance. Instead, each of the 22 gene trees was built using homologs from a single gene class. This means that genes in each tree have a shared common ancestor. We have clarified this part of the text (Line 541).

The 22 trees correspond to gene models from fARGene, which uses hidden Markov models to identify resistance genes in bacterial genomes. Each gene model is built from a set of ARGs that share an evolutionary history and induce a similar resistance phenotype (previously described in detail in Berglund et al. 2019, cited in the paper). These ARGs have been manually selected and curated from the literature, and fARGene optimizes each model against a set of 'negative sequences', i.e., genes that also share an evolutionary history but does not induce any known resistance phenotype. The 22 included hidden Markov models are, thus, very specific regarding what genes they identify. For example, class B1/B2 and B3 beta-lactamases are identified by separate models (even though they are homologs) since they do not have a common ancestor that induce a resistance phenotype (since the resistance mechanisms have developed in parallel). AAC(3) aminoglycoside acetyltransferases are also identified by two separate models even though they act through the same resistance mechanisms since these enzymes are encoded by two types of genes that are not homologous. The high accuracy of fARGene has also been confirmed previously by experimental validation (e.g., Lund et al. 2023, Lund et al. 2022, Berglund et al. 2020).

Indeed, throughout the manuscript, improvements to the text are necessary to improve understandability. I also include methodological questions. Comments in order of appearance:

1. *Comment:* Line 73 – Typo, 2.6 million

Reply: The typo has been corrected.

2. *Comment:* Line 80 – The number of metagenomes associated with some microbiomes (e.g. human and wastewater) are likely to outnumber many other types of communities by the nature of the types of metagenomic sequencing that we tend to do. This should be included as a caveat of this work. I do see it briefly mentioned at line 312 but a more thorough discussion/investigation is needed.

Reply: We agree that when looking at the total amount of metagenomic sequencing data that is available today, it is likely that environments like the human gut are overrepresented in terms of the number of samples. To reduce this bias, we based the study on two major dataset: 1) the Earth Microbiome Project (EMP), where the number of samples from human microbiomes (3,244) is actually lower than samples from animals (4,325), soil (4,137), and water (7,898) and 2) the Global Water Microbiome Consortium database (1,187 samples). Thus, human and wastewater samples do not outnumber samples from other environments in our dataset.

It should, furthermore, be noted that we never directly count the number of samples where a bacterial host could be detected. Instead, we estimate the co-occurrence as the proportion of samples from each environment where OTUs representing the genomes of both sides of a transfer could be detected. Also, we only considered relative changes, i.e., differences in co-occurrence between observed and random transfers, which means that changes in low frequencies may be of interest. In addition, we apply a conservative threshold (presence in at least 1% of the samples), which prevents spurious matches and hosts from being repeatedly detected also in environments with a low number of samples.

Nevertheless, we agree with the reviewer that this is an important aspect and have included the number of metagenomic samples from each category to the Methods section (Line 629).

3. *Comment:* Figure 2 – Acronyms used in the Figure need to be included in the legend (e.g., ARG, Resistance mechanisms). A brief summary of the resistance mechanisms should be included in the text (Introduction or appropriate Results section) to put the Results in context. (And in all Figures/Sup Figs).

Reply: This would, indeed, improve the clarity, and we have added explanations of all

relevant acronyms in the figure legends, as well as a brief description of the relevant resistance mechanisms where they are introduced in the Results section (Line 93).

4. *Comment:* Figure 2d – y axis is missing a title.

Reply: A title (“Performance”) has now been added to the y-axis of Fig. 2d.

5. *Comment:* Line 161/Fig4ab- I don’t understand what I’m looking at here, and I suspect most of your desired readership won’t either. A few sentences explaining partial dependence analysis and walking your audience through your analysis is needed here. Why were no other factors looked at in this way? Only genome-gene 5mr and mean 5mr.

Reply: These plots show the result from the partial dependency analysis – a commonly used approach to investigate how continuous factors influence the predictive power of a model. We do agree that this form of analysis may not be well-known to all readers and have therefore included a short explanation (Line 221).

We did not include this analysis for any of the nominal categorical features, such as Gram staining, since it does not provide any useful visualization. The co-occurrences, which are always defined between pairs of hosts, were instead visualized using networks, which more clearly reflect its graph structure. Finally, we did not include this analysis for the genome size difference due to the low importance of this factor in all models.

6. *Comment:* Sup Fig 9 missing?

Reply: We had accidentally labeled Sup Fig 9 as Fig 3 in the Supplementary Material. This has now been corrected.

7. *Comment:* Line 212-3 – Was this quantified? “Compared to the human microbiome, the co-occurrence of host-pairs in wastewater samples was, overall, higher.”

Reply: Since we estimated the co-occurrence as the proportion of samples from each category where host-pairs were simultaneously detected, yes. We have rephrased this sentence to make this more clear (Line 297).

8. *Comment:* Sup Fig 10d- Is this across all samples?

Reply: Panel d in Sup Fig 10 shows connections between taxa which were observed together ≥ 5 times in our identified ARG transfers. As such, no information from the metagenomic samples is included here, instead it serves as a reference for all of the potential connections that we could have observed in the co-occurrence networks shown in Fig 5, as well as Sup Fig 10 a,b,c. This has been clarified in the figure legend.

9. *Comment:* Sup Fig 11- Legend does not match the mention of this Figure in the text. What is being represented? How does this differ from Sup Fig 10?

Reply: We agree that the mention of Sup Fig 11 in the text could be interpreted in a way that was misleading. Sup Fig 11 is different from Sup Fig 10 in both how the nodes have been aggregated and how the edges have been drawn. In Sup Fig 10, the aggregation is the same as in Fig 5, i.e., a selection of important pathogens was retained as individual nodes while other species were aggregated to a higher taxonomic level. The edges represent all observed transfers with sufficient co-occurrence between the nodes. In Sup Fig 11, the species that were not aggregated were those that were identified as recent origins of mobile ARGs in Ebmeyer et al 2021 (<https://doi.org/10.1038/s42003-020-01545-5>). Here, the only edges that are drawn are those representing frequent transfers between these origin species and other taxa, including connections where little to no co-occurrence was estimated (which are represented as dashed lines). We have attempted to clarify the legend of Sup Fig 11, as well as in the manuscript (Line 338). Considering the complexity of the network, we argue that both these figures have merit, where Sup Fig 11 may be of special interest for those interested in transfers involving species that have been identified as recent origins of ARGs.

10. *Comment:* Line 308- Is this new data being introduced in the Discussion?

Reply: This data is inferred from Fig. 5, so it is not new per se. However, we agree that it would be good to mention these numbers in the Results section as well, and have added them (Line 299).

11. *Comment:* Line beginning 314- I don't understand how the example provided (M. morganii and S. algae) corresponds to the point of recent origins of ARGs?

Reply: M. morganii and S. algae are both identified as recent origins of mobile ARGs (DHA and QnrA, respectively) in Ebmeyer et al 2021. Here, they merely serve as examples to illustrate that these origin species (like other promiscuous bacteria) were more likely to co-occur with the bacteria that they were observed to transfer ARGs within the human microbiome than they were in wastewater. This has been clarified in the text (Line 338).

12. *Comment:* Line 355- Typo, ARGs.

Reply: The typo has been corrected.

13. *Comment:* Line 363- How many genomes were downloaded? How many genomes were there before and after curation? How was "where contamination was suspected" determined?

Reply: In total, 1,150,159 genomes were downloaded. During curation, 282,841

genomes were removed, leaving 867,318 genomes for the analysis. Like the information regarding the reliability of the taxonomic annotation, metadata from NCBI also contains information about suspected contamination of genome assemblies, and this metadata was used to remove genomes where contamination was suspected. These details have now been added to the manuscript (Line 512).

14. *Comment:* Line 370 – Typo, ARG.

Reply: This has been rephrased.

15. *Comment:* Line 370 – “predicted translated ARGs were clustered at 100% amino acid identity” what does this mean here? To me, that reads as individual ARG homologs were determined from the results of fARGene by clustering for similarity at 100%; however, that doesn’t really make sense with the next steps of MSA alignment, tree building. How were individual ARG gene clusters identified here?

Reply: We clustered the predicted protein sequences from each gene class at 100% similarity to remove redundancy. The main reason for clustering is computational since it proved infeasible to build trees from all predicted instances of the larger gene classes. Redundant sequences can also bias the construction of the phylogenetic trees. We have clarified this part of the Methods section (Line 549).

16. *Comment:* Line 374- What is the distribution of identified ARG gene clusters/individual protein types? Does Sup Fig 1a, for eg, include a single ARG homologous gene cluster? If not, how many homologs per phylogeny and what justifies including them in a single gene tree?

Reply: The numbers of unique protein sequences per gene class from which each tree was constructed are displayed in Supplementary Table 1. As described above, each fARGene model has been optimized to identify resistance genes that share an evolutionary relationship, which justifies using the sequences predicted by each model to create phylogenetic trees (one for each fARGene model/resistance gene class), which are displayed in Sup Fig 1.

17. *Comment:* Line 414- Have the authors considered the implications of focussing on more environmental-type datasets? For e.g. given the focus on human microbiomes in the text, I am surprised that metagenomic data from the Human Microbiome Project was not included?

Reply: The reason that we chose to include the Earth Microbiome Project (EMP) is specifically because it is very comprehensive and consistently generated. We needed, however, to complement with additional data representing wastewater since this was missing from the EMP, but since the EMP did include a lot of samples from the human

microbiome (3,244 samples) we saw no need to add additional human samples.

18. *Comment:* Line 419- How were cases where an OTU matched multiple genomes of different taxonomic assignments dealt with? OTUs are short and often not specific to a given genome, strain, or even species. Why wasn't a widely used approach for OTU taxonomic calling used here?

Reply: When mapping genomes to OTUs, we did not consider the taxonomy of the genomes. Instead, for each genome, we found the best match based on sequence identity, provided that it passed the thresholds of >97% sequence identity and >90% coverage, which are widely used alignment criteria for OTUs. We agree with the reviewer that OTUs are not specific, and in our analysis we did not limit the OTUs to one matching genome each, we only limited genomes to one matching OTU each. That is to say, we allowed multiple genomes to match the same OTU. This has been clarified in the text (Line 622).

19. *Comment:* Line 426- How was co-occurrence determined? No thresholds or analyses tools/methods mentioned.

Reply: We did not use any third-party tools to estimate the co-occurrence. For each observation and environment type, we first generated all possible combinations of OTUs represented on the different sides of the transfer. Then, for each combination of OTUs we calculated the proportion of samples where both OTUs were present (≥ 3 reads). This has been clarified in the manuscript (Line 633).

20. *Comment:* Line 446- How many were removed?

Reply: In total, 3,419 transfers were removed before creating the models (3,409 of which were discarded because of missing OTU data). This number has been added to the manuscript (Line 657), along with a correction of the number of transfers that were removed due to missing OTUs (Line 625).

21. *Comment:* Figure 2 – Has any correction been made for more commonly sequenced organisms? For e.g. if an HGT from E.coli to Staph aureus was identified 100x but an HGT between 2 less-well studied organisms was identified only 10x, was some sort of normalization performed?

Reply: We are well aware of the inherent bias in sequence repositories. All our comparisons were therefore made in relation to a null model, where transfers were randomized between bacterial hosts (described at Line 663). This provides an estimate that describes the influence of the factors conditional on the species distribution in the database. It should, however, be mentioned that it is impossible to compensate for highly undersampled species, for which horizontal gene transfers are rare or completely absent in existing databases. In fact, we argue that the main issue is not the oversampling of

common pathogens but rather the undersampling of environmental and commensal bacterial hosts that may serve as important hosts in the dissemination of ARGs (which is also raised by reviewer 3). We have further emphasized this limitation of the data in the Discussion (Line 468).

22. *Comment:* Sup Fig 1- Threshold for the number of times a HGT was seen to call it a HGT? For e.g. if the ARG is identified in a phyla 100 times and in a second phyla once, will that be counted?

Reply: We did not implement any such threshold here. In part, this is due to the aforementioned bias in the databases, where many transfers between commonly sequenced bacteria and rare bacteria would be overlooked if such a threshold was implemented. We have clarified this in the manuscript (Line 560). We did, however, omit genomes with unreliable taxonomic annotations, as described on Line 513, to improve the robustness of our analysis. Furthermore, since each model was trained on a large number of transfers, instances like the ones you describe are unlikely to have had a significant contribution to the results.

23. *Comment:* Fig 3a- The Results section is missing an explanation of which factors were taken into account in the ML RF model: what they are, how they were calculated, why they were included over other factors? For example, the sentence starting at line 143 and pointing to Fig 3b discusses genome/gene differences in nucleotide composition; is this the genome-gene 5mer distance data in Fig 3b? I appreciate that this is included in the Methods but, as written, this isn't seen to the reader until after the Results are presented- some of this could be rectified by changing the axis labels.

Reply: We agree that including more information about the selected factors in the Results section would help with improving the clarity of the manuscript. A brief description of each factor has now been included (Line 138).

24. *Comment:* Line 411- First mention of genome size in the text; would be helpful to bring this into the Results section.

Reply: In addition to the new explanation of the included factors, we have added an additional mention of the genome size to the Results section (Line 198).

25. *Comment:* Discussion of the impact of fragmented MAGs might have on this work?

Reply: Of the 867,318 genomes that were screened during our analysis, only 6,449 (0.74%) were annotated as being derived from metagenomes. These are unlikely to have influenced the results to any significant extent. We have added this number to the Methods section (Line 516) but we argue that it does not warrant any longer discussion

in the manuscript.

26. *Comment:* In the Introduction, the authors outline how genes are HGTed including that many genes will exist on extra chromosomal elements. What was the distribution of ARGs on extra-chromosomal elements vs. chromosomes? In the metagenomic data, how were extra-chromosomal elements linked to chromosomes (and organisms)?

Reply: The metagenomic data used in this study only encompassed 16S amplicon data. Therefore, this data could only be used to estimate the presence of different bacteria in various environments, and not the presence of different genes (such as those encoding mobile genetic elements). In the genomic data, the presence of MGEs is difficult to assess since most of the genomes are not fully assembled. Furthermore, while many mobile genetic elements, in particular in Pseudomonadota, have been well-characterized, much is still unknown about the MGEs used by other bacterial phyla – this makes such analyses inconclusive. Furthermore, the distribution of ARGs on extra-chromosomal elements has recently been analyzed by other studies (see, e.g., Che et al., Conjugative plasmids interact with insertion sequences to shape the horizontal transfer of antimicrobial resistance genes, PNAS 118 (6) 2021).

Reviewer 3

This interesting work is based on the application of a machine learning strategy leveraging over 2.6 million ARGs identified in almost 1 million genomes and predicting horizontal gene transfer of antibiotic resistance genes (ARGs). The work, using an extensive metagenomic analysis, also explores ecological connectivity and the phylogenetic neighborhood and consequently predicts horizontal ARG transfer between bacterial hosts. The results of such huge research do not provide really novel results or suggestions, but certainly strongly reinforce previous hypotheses and findings. A weakness of the manuscript is the inherent bias of databases, with an anthropocentric composition of the analyzed microorganisms, preventing a generalization of the results to the microbiosphere. However, the data provided are certainly of interest for future research in the field, and in my opinion, the manuscript deserves publication, with some additions/discussions that I am mentioning below.

1. *Comment:* Line 52. This referee agrees that the transmission of genes from environmental/commensal organisms to pathogens is a critical question in AMR research.

Reply: We thank the reviewer for the constructive comments, and are happy to hear that the importance of ARG transmission into pathogens is recognized.

2. *Comment:* Line 58: the authors, after mentioning transformation, should add the "new mechanism" of AMR transfer by extracellular vesicles. See, for instance, Qu S, Zhang Y, Weng L, Shan X, Cheng P, Li Q, Li L. The role of bacterial extracellular vesicles in promoting antibiotic resistance. Crit Rev Microbiol. 2024 Nov 4:1-18. doi:

10.1080/1040841X.2024.2423159

Reply: We agree that this mechanism is also likely to play a role and have added this to the manuscript (Line 57).

- Comment:* Line 67. This is a hot point in our days. Add the recent reference: Akob DM, Oates AE, Girguis PR, Badgley BD, Cooper VS, Poretsky RS, Tierney BT, Litchman E, Whitaker RJ, Whiteson KL, Metcalf CJE; Ecology Evolutionary and Biodiversity Retreat Participants. Perspectives on the future of ecology, evolution, and biodiversity from the Council on Microbial Sciences of the American Society for Microbiology. mSphere. 2024 Nov 21;9(11):e0030724. doi: 10.1128/msphere.00307-24.

Reply: We agree that this is a hot topic and have included the suggested reference.

- Comment:* Line 81. The authors should acknowledge that the available repositories are predominantly populated with pathogenic, clinical, and sewage isolates, thereby introducing a bias that should be addressed for the sake of objectivity. The most viable solution is to encourage future research to quantify (or at least estimate) the proportion of different microorganisms in the various ecological compartments/patches and normalize the results accordingly.

Reply: We are well aware of the biases in the databases. All our comparisons were made in relation to a null model, where transfers were randomized between bacterial hosts (described in Line 663). This provides estimates that are conditional on the species distribution in the database. We argue, similarly to the reviewer, that the main issue is not the oversampling of common pathogens but rather the undersampling of environmental and commensal bacteria that may serve as important hosts in the dissemination of ARGs. We have included acknowledgments of this bias in the Discussion section (Line 420, Line 468).

- Comment:* Line 147. That is an oversimplification. Gram staining does not mean a different cell wall (muropeptide) composition, in any case, a different "cell envelope". Note that many Gram-negative groups (as Negativicutes) are phylogenetically close with others that are Gram-positive by staining. It could be nice to see how these "pseudo-Gram negatives" behave with other "true" Gram negatives. In any case that should be discussed.

Reply: As per the reviewers suggestion, we have changed "cell wall composition" to "cell envelope" throughout the manuscript.

We use Gram-staining as a proxy for the differences in the bacterial cell envelope observed among bacteria (e.g. monoderms and diderms). Unfortunately, there is no comprehensive database describing this property for bacteria, which is the reason we

apply "consensus gram staining". This is far from perfect: Indeed, Bacillota in particular contains some Gram-negative taxa, although it is predominantly comprised of Gram-positives. Examples of taxa where this assumption does not hold include the class Negativicutes, mentioned by the reviewer, and the order Eubacteriales/Clostridiales. Given the lack of structured bacterial morphology data and the scope of this study, comprising almost 1 million bacterial genomes, we argue that this is an acceptable approximation. Indeed, we estimate that our approach has resulted in incorrect labeling only for a small part of the included horizontal transfers. In particular, only ~3% of the transfers involved Negativicutes. Even if this has a minor effect on the overall performance, we cannot rule out that the model may have a lower accuracy when predicting transfers involving bacteria that do not follow our consensus classification. This is now mentioned in the Discussion (Line 482).

6. *Comment:* Line 151. The authors should make clear in their message that the key aspect of the dissemination of ARGs by horizontal transfer depends on the promiscuity of the mobile genetic elements carrying ARG genes.

Reply: We agree that mobile genetic elements play a central role in the dissemination of mobile ARGs, even though they are not the main focus of our study. We have now made this clear in the manuscript (Line 192).

7. *Comment:* Line 158. That reflects the over-representation of human and animal strains in databases, and not necessarily the reality in nature. See my previous comment.

Reply: This is, as mentioned above, now acknowledged in the Discussion section (Line 420, Line 468).

8. *Comment:* Line 182. This point cannot be presented here as a "discovery", as it was suggested decades ago in the microbiological literature (part of it can be found for instance for sure in in your reference 3)

Reply: We have rephrased this to better reflect the contribution of our results (Line 256), which is the relative contribution of these effects based on large-scale genomic analyses.

9. *Comment:* Line 192. Again, that should be considered in light of the promiscuity of mobile genetic elements.

Reply: We agree that MGEs are relevant to these results as well, and have now included this in the manuscript (Line 269).

10. *Comment:* Line 225. This referee agrees with the fact that connectivity and niche neighborhoods facilitate ARG transmission. An important point to consider (and eventually discuss in the Discussion section) is the "coalescence" or "merging of

microbiomes" of different origins in microbiotic particles (Baquero, F., Coque, T. M., Guerra-Pinto, N., Galán, J. C., Jiménez-Lalana, D., Tamames, J., & Pedrós-Alió, C. (2022). The influence of coalescent microbiotic particles from water and soil on the evolution and spread of antimicrobial resistance. *Frontiers in Environmental Science*, 10, 824963.). The co-occurrence of organisms of disparate origins can be explained in such a way.

Reply: We thank the reviewer for this comment. We have now included this in the Discussion section (Line 448).

11. *Comment:* Line 247. Of course, not only ARGs are submitted to horizontal gene transfer. That is why in the work referenced (21) the study included all genes from the "accessory genome" (ARGs are part of this genome). Mobile genetic elements able to transfer non-ARG traits can be exploited to disseminate ARGs if they are able to capture (from the chromosome or from other mobile genetic elements) these antibiotic resistance genes.

Reply: This is, indeed, true. However, we focus on ARGs in this study, and are, therefore, not in a position to draw any conclusions about the horizontal transfer of other genes.

12. *Comment:* Lines 254 or 267. However, translational demand is not influencing plasmid-associated fitness costs (Rodríguez-Beltrán, J., León-Sampedro, R., Ramiro-Martínez, P., de la Vega, C., Baquero, F., Levin, B. R., & San Millán, Á. (2022). Translational demand is not a major source of plasmid-associated fitness costs. *Philosophical Transactions of the Royal Society B*, 377(1842), 20200463).

Reply: We thank the reviewer for highlighting this. A mention of this has been included in the Discussion (Line 365). It should, however, be pointed out that the study provided by the reviewer is limited to plasmids in Enterobacteriaceae and short-term experiments (24h). In contrast, our results are based on transfers at the order level and above and over much longer time spans, which may affect the importance of translation efficiency in new hosts.

13. *Comment:* Line 274. That is what I mentioned above.

Reply: As stated in the reply to a previous comment, we agree that this in itself is not a novel finding.

14. *Comment:* Line 323. That is discussed in detail in Hernando-Amado, S., Coque, T. M., Baquero, F., & Martínez, J. L. (2019). Defining and combating antibiotic resistance from One Health and Global Health perspectives. *Nature microbiology*, 4(9), 1432-1442.

Reply: We have added the suggested reference to the manuscript.